# Electrocatalytic Reduction of CO_2_ in Water by a Palladium-Containing Metallopolymer

**DOI:** 10.3390/nano12071193

**Published:** 2022-04-02

**Authors:** Marcos F. S. Teixeira, André Olean-Oliveira, Fernanda C. Anastácio, Diego N. David-Parra, Celso X. Cardoso

**Affiliations:** 1Department of Chemistry and Biochemistry, School of Science and Technology, Sao Paulo State University (UNESP), Presidente Prudente CEP 19060-900, SP, Brazil; andre.olean-oliveira@unesp.br (A.O.-O.); funcao@gmail.com (F.C.A.); david.parra@unesp.br (D.N.D.-P.); 2Department of Physics, School of Science and Technology, Sao Paulo State University (UNESP), Presidente Prudente CEP 19060-900, SP, Brazil; xavier.cardoso@unesp.br

**Keywords:** metallopolymer, electroreduction of dioxide carbon, electrocatalysis, aqueous system, overpotential

## Abstract

The palladium–salen complex was immobilized by electropolymerization onto a Pt disc electrode and applied as an electrocatalyst for the reduction of CO_2_ in an aqueous solution. Linear sweep voltammetry measurements and rotating disk experiments were carried out to study the electrochemical reduction of carbon dioxide. The onset overpotential for carbon dioxide reduction was approximately −0.22 V vs. NHE on the poly-Pd(salen) modified electrode. In addition, by combining the electrochemical study with a kinetic study, the rate-determining step of the electrochemical CO_2_ reduction reaction (CO_2_RR) was found to be the radial reduction of carbon dioxide to the CO adsorbed on the metal.

## 1. Introduction

The use of fossil fuels as the main source of energy in the last century has drastically increased levels of CO_2_, one of the main gases responsible for the greenhouse effect [1,2]. The electrochemical reduction of CO_2_ in an aqueous solution is a promising solution to the problem, and this method can be used in a direct fuel cell or to convert CO_2_ into low molecular weight compounds such as CH_4_, CO, HCOOH, and CH_3_OH [3,4,5,6].

Several materials have been extensively explored to develop a suitable electrocatalyst for CO_2_ reduction [6]. These materials include organic molecules, ionic liquids, metal complexes, metal salts, nanoparticles, and polymers [7]. The great challenge for catalysis using these metals is to increase the faradaic efficiency, diminish the overpotential of catalysis, and ensure that the material is not poisoned by the formed byproducts [7,8,9,10]. Huang and collaborators [1] used a Pd icosahedra/C electrode and reached a faradaic catalytic efficiency of 91.1% for the selective conversion of CO_2_ to CO at a potential of −0.80 V vs. RHE. In another work, Kortlever et al. [3] used carbon-supported bimetallic nanoparticles to create a Pt–Pd electrode and reached a faradaic efficiency of 88% for the conversion of CO_2_ to formic acid with a low overpotential (−0.40 V vs. RHE). These works demonstrate the catalytic potential of Pd-based materials in the search for better catalytic efficiencies and reduced catalytic overpotentials.

In the recent past, metallopolymers attracted much attention because of their excellent combination of redox-mediated switching capabilities, conductivity, and electrocatalytic properties [11,12,13,14,15]. According to Elmas et al. [16], metallopolymers exhibit a well-defined coordination environment, which can be applied in the development of smart materials with the redox properties of a transition metal cation. In electrocatalytic applications, metallopolymers function as hybrid materials capable of incorporating better catalytic properties than transition metals in conducting systems [17,18,19,20,21]. In our research group, we have studied the electrocatalytic properties of metallopolymers containing salen moieties complexed with different metals in the application of fuel cells [21,22], chemiresistor sensors [23], and electrochemical sensors [24,25,26]. The materials consist of an extended three-dimensional network formed by axial coordination of the d orbitals of the central metal with the π orbital of the phenolate moiety of the adjacent complex. Of these contributions, we have investigated in detail the effects of the different redox-active and charge transport properties of the metallopolymers [23,27,28,29].

In this study, we reported the catalytic activity for the electrochemical reduction of carbon dioxide in an aqueous medium at room temperature using a platinum-modified electrode with poly-Pd(salen). The metallopolymer shows a distinctive overpotential reduction of carbon dioxide in a slightly acidic solution. In addition, kinetic studies of RDE and turnover frequency are discussed.

## 2. Experimental

### 2.1. Synthesis of Pd(II) 2,2-{1,2-Ethanediylbis[nitrilo(E)methylylidene]}diphenolate

The ligand 2,2-{1,2-ethanediylbis[nitrilo(E)methylylidene]}diphenolate (salen) was purchased from Sigma Aldrich (St. Louis, MI, USA) and used without further purification. The metallic complex [Pd(salen)] was prepared by the addition of stoichiometric and equimolar amounts of the *Schiff* base ligand and palladium acetate (Sigma Aldrich, St. Loius, MI, USA) in absolute ethanol. The solution was kept under reflux for 3 h at 50 °C with constant stirring to yield a yellow precipitate. The precipitate was filtered in a Gooch crucible, washed with absolute ethanol, and stored in a desiccator.

### 2.2. Electrosynthesis of Metallopolymer on Platinum Electrode

The metallopolymer was formed by electropolymerization using 1.0 mmol L^−1^ [Pd(salen)] complex in acetonitrile (Sigma Aldrich, St. Loius, MI, USA, 99.9%) with tetrabutylammonium hexafluorophosphate (TBAHFP) and tetrabutylammonium perchlorate (TBAPC) (Sigma Aldrich, St. Loius, MI, USA) electrolyte at a concentration of 0.10 mol L^−1^ while applying a potential range of 0.0 to +1.2 V vs. SCE at 100 mV s^−1^ in a N_2_ atmosphere. Electropolymerization was performed in a conventional electrochemical cell with three electrodes: a saturated calomel electrode (SCE) as the reference electrode, a platinum wire electrode as the auxiliary electrode, and a platinum electrode (A = 0.071 cm^2^) as the conductor substrate for coating with metallopolymer.

### 2.3. Morphological Characterization

The morphological features of poly-Pd(salen) were analyzed by atomic force microscopy (AFM) and scanning electron microscopy (SEM). Atomic force microscopy images of the metallopolymer were captured with a Nanosurf (Boston, MA, USA) EasyScan 2 AFM. The scan head was imaged at an angle of 45 degrees relative to the axis of stretching. The collected images were analyzed with Nanosurf AFM software (version 3.0.2.4) to determine roughness and thickness. SEM was performed using an EVO 50EP (Carl Zeiss SMT AG, Oberkochen, Germany) scanning electron microscope (SEM). All observations were carried out with a secondary electron (SE) detector in high-vacuum mode at an accelerating voltage of 15 kV. All morphological analyses were performed on a platinum-coated metallopolymer.

### 2.4. Electrochemical CO_2_ Reduction Reaction on Electrode Modified with Metallopolymer

Electrochemical measurements were carried out on a potentiostat/galvanostat μ-Autolab type III (Metrohm Autolab, Utrecht, The Netherlands). Carbon dioxide reduction on the modified electrode was carried out in a standard three-electrode cell, with an SCE reference electrode and a platinum wire electrode as the counter electrode. The modified electrode was tested in 0.5 mol L^−1^ KCl (pH 4–7) aqueous solution. Initially, the solution was continually purged with N_2_ to remove the dissolved oxygen from the solution. Then, the solution was purged continuously with carbon dioxide, while the solution was analyzed electrochemically at a rate scan of 25 mV s^−1^. The solution saturated with carbon dioxide had a pH of 4.7. Rotating disk electrode voltammetry (RDEV) was conducted with a μ-Autolab type III (Metrohm Autolab, Utrecht, The Netherlands) connected to a microcomputer and controlled by GPES software. RDE measurements were performed with a motor speed controller (Autolab RDE, Metrohm Autolab, Utrecht, The Netherlands). The platinum electrode modified with metallopolymer (diameter 3.0 mm) was rotated between 400 and 4000 rpm using a CO_2_ saturated 0.5 mol L^−1^ KCl solution.

## 3. Results and Discussion

Figure 1 shows the cyclic voltammograms for the electropolymerization of the metallopolymer film on the conductor platinum substrate. The application of a positive potential limit of +1.0 V (vs. SCE) is required for the formation of radical cations, which initiate the polymerization process [22,30]. The first potential cycle (red line) resulted in only one redox process during the anodic scan at a potential close to 1.0 V; this process is associated with the cation radicals. With the increase in scan number, a redox system began to be evident in the region of 0.40 to 0.80 V (vs. SCE), indicating an increase in electroactive material on the surface of the platinum electrode. The redox system is attributed to the Pd^II^/Pd^III^ redox activity of the generated polymeric film.

The typical electron-forming mechanism of films based on metal–salen type complexes has been extensively studied and consolidated in the literature. The electron-deficient cation radical formed during the anode scan interacts with the electron-rich π-conjugated system of an adjacent monomer, forming bridges between the molecules [21,25,26,31]. This mechanism is possible due to the effective interactions between the d-center orbitals of the metal center and the π orbitals of the aromatic ring of a second metal-complex molecule. This interaction allows the formation of a backbone coupled with an electron-hopping-based transport mechanism [29].

The electrode coating with the catalytic material was confirmed by AFM characterization. Figure 2A,B show three-dimensional surface topography images of the uncoated and coated electrode surfaces, respectively. 

The images were taken over a 100 μm^2^ scanned area. Due to the formation of the film by the center metal/aromatic ring interaction, metal–salen metallopolymers typically present stacked column morphologies with irregular topographies. The root mean square (RMS) roughness of the platinum surface was 28.0 nm, which increased to 57.6 nm following the formation of the metallopolymer. There is also an apparent distinction between the surface morphology of electrodes, where the image of the coated surface (Figure 2B) exhibits a visible nanogranular structure. Atomic resolution images with four lines and cross-section profiles of the uncoated and coated surfaces are shown in Figure 2C,D, respectively. Height profiles along the lines revealed notable differences in the surface features of metallopolymers. In general, metallopolymer columns were observed to exhibit 1.39 or 2.4 μm periodicity with random distribution across the film, and the granular height range was 0.09–0.38 μm. The uncoated platinum in the image (Figure 2C) does not exhibit the same periodic surface characteristics, and the height profiles are much smaller. This observation might be interpreted as the influence of the counteranion size of the supporting electrolyte during the electropolymerization of the palladium complex. The distribution of the particles over the electrode surface is controlled by the radius ion of the hexafluorophosphate anion [32]. During oxidative polymerization, the molecular columns acquire a positive charge and are normally arranged on the electrode surface at distances that are determined by the diameter of the anions and solvent molecules. These results of the metallopolymer morphology by AFM are in close agreement with the analysis obtained by SEM data, which is reported in a later section. Figure 3 presents an SEM image of the poly-Pd(salen)-modified electrode, and the image illustrates that the metallopolymer was formed by randomly stacked columns. Quantitative EDS microanalysis revealed that the elemental surface composition of the electrode was composed of Pd (2.5%), C (50.6%), O (22.0%), and O (24.1%). The low percentage of platinum obtained in the microanalysis indicates the complete coating of the electrode surface by the metallopolymer. This observation ensures that the CO_2_ electroreduction studies have no significant influence on the platinum catalytic activity.

### Electrochemical Reduction of Carbon Dioxide Catalyzed by a Pd(salen) Metallopolymer

To verify the electrocatalytic activity of the metallopolymer for carbon dioxide electrochemical reduction in an aqueous medium, cathodic linear sweep voltammetry experiments were performed in a CO_2_ saturated solution with platinum and metallopolymer-modified platinum electrodes (Figure 4). The onset overpotentials for dioxide carbon reduction were approximately −0.74 V and −0.46 V vs. SCE on the platinum electrode and poly-Pd(salen)-modified electrode, respectively. Based on the experimental results, we can see that the cathodic current increases for the modified electrode, requiring 0.15 V less overpotential than the platinum electrode to carbon dioxide electroreduction when considering the isocurrent for both electrodes. This decrease in overpotential is crucial for the effective electrochemical reduction of CO_2_ by the catalyst. Moreover, the catalytic current of the modified electrode was 3.3 times greater than that of the unmodified platinum electrode at an isopotential of −0.95 V vs. SCE.

To confirm the electrocatalytic response of the electrode modified to dissolved carbon dioxide, the metallopolymer platform was tested in the absence of carbon dioxide (dashed curve of Figure 4). In the absence of carbon dioxide in the electrolytic solution, the metallopolymer-modified electrode showed two reduction peaks (−0.56 V and −0.80 V) related to the transformation of Pd(II) to Pd(I) and Pd(I) to Pd(0), respectively. Under a carbon dioxide atmosphere (orange curve), the current increased significantly after a potential of −0.70 V was applied, clearly indicating that the total reduction of the metal center drives the overall reduction of dissolved carbon dioxide.

The anionic counterion used in the electropolymerization process of metal–salen complexes affects the electronic properties and physico-chemical characteristics of the originally formed metallopolymer [25,29,32]. The counteranion used in the electropolymerization process has a direct influence on stability during the formation of the radical cation and may, consequently, alter the network of the generated polymer and its electrochemical performance. For redox-conducting polymers based on the metal–salen complex, the counteranion size directly affects the stability and favors a more orderly and regular polymer structure. For this study, two modified electrodes were prepared using hexafluorophosphate (Ø = 0.87 nm) and perchlorate (Ø = 0.48 nm) as counteranions in the electrolyte supporting the electropolymerization reaction. The electrochemical performance of these devices in the catalytic reduction of dissolved carbon dioxide was determined by cathodic linear sweep voltammetry in a KCl solution. In Figure 5A, the device prepared with hexafluorophosphate has a higher current magnitude for carbon dioxide electrocatalysis than the polymer obtained with perchlorate. In a comparative analysis of the polarization curves (Figure 5B), three slopes are observed in both studied electrodes, clearly indicating the steps involved in the electroreduction of dissolved carbon dioxide. Tafel analysis reveals that the Tafel value of the metallopolymer prepared with hexafluorophosphate was much lower in the first stage (region 1), which represents the reductive adsorption of carbon dioxide by the central metal. The counteranion imposes size and shape selection restrictions through molecular channels readily adjusted in the polymer, exhibiting a molecular sieving effect. In comparison, the hexafluorophosphate anion introduced molecular spaces of the appropriate size into the polymer to stabilize carbon dioxide, improving the efficiency of the coupling reaction. This first stage is crucial for the efficiency of the catalysis but is not the rate-determining step. Subsequently, regions 2 and 3 are ascribed to carbon dioxide radial (CO_2_^•−^) reduction to adsorbed CO and -CHO, respectively [33,34,35]. Figure 1 presents a proposal for the steps involved in the electrochemical CO_2_RR.

The Tafel slopes for these polarization regions were similar for both electrodes, and region 2 was the rate-determining step for carbon dioxide reduction. Palladium complex catalysts show trends for the formation of CO and HCOOH [36,37]. The next electrocatalytic studies were carried out with a metallopolymer-modified electrode obtained by tetrabutylammonium hexafluorophosphate. The effect of pH on the polarization curve of the CO_2_RR was carried out in an electrolyte saturated with carbon dioxide. The potentials as a function of pH were monitored at two current densities (2.0 and 5.0 mA cm^−2^) corresponding to regions 2 and 3 of the electroreduction process. The plot of the applied potential versus pH (see Appendix A) showed slopes of 46 mV pH^−1^ (r = 0.9914) and 45 mV pH^−1^ (r = 0.9978) for regions 2 and 3, respectively. This study is consistent with the theoretical value for an electrochemical CO_2_ reduction process involving one electron and one proton in each step.

Rotating disk electrode measurements were performed to gain insight into the CO_2_RR performance of the poly-Pd(salen)-modified electrode. Figure 6A shows the linear sweep voltammogram for various rotation rates of 400 to 4000 rpm at a scan rate of 10 mV s^−1^. The cathodic current increased in magnitude with increasing rotation rate. Figure 6B represents the relationship between the cathodic current and the square root of rotation (Levich plots) at three fixed potentials corresponding to the regions presented in the Tafel study. In the Levich analysis at −0.75 V, the current as a function of the square root of the rotation was linear (the black circle in Figure 6B), indicating diffusion-controlled carbon dioxide reduction.

At potentials of −0.85 V and −1.05 V, the current as a function of the square of the rotation rate showed nonlinearity. For these potentials, there is a dependence on the value of the catalytic current at low rotations, but for rotations above 1000 rpm (104.7 rad s−^1^), the catalytic current is not significantly influenced. This behavior demonstrates that the catalytic current is limited by a slow chemical step, as noted earlier in the Tafel study. For further analysis, the data obtained were applied to the Koutecky–Levich equation [38], and the linearity at −0.85 V and −1.05 V (see Figure 6C) confirms the pseudo-first-order kinetic behavior with respect to the concentration of adsorbed carbon dioxide. However, the kinetic parameters (such as the heterogeneous electron transfer rate constant) were not determined due to the difficulty in determining the diffusion-limited current densities at each stage.

Based on previous studies, a rough catalytic turnover frequency (TOF) estimate was determined using the following equation [39]:(1)$$\mathrm{TOF}=\frac{j}{nF\Gamma }$$
where *j* is the current density (A cm^−2^) at an applied potential (1.20 V vs. SCE), *n* is the total number of electrons involved in the reaction (CO_2_RR = 3e^−^), *F* is the Faraday constant (96 485 A s mol^−1^), and *Γ* is the surface-active site concentration (6.0 × 10^−10^ mol cm^−2^) calculated by integrating the current in the CV [40]. The modified electrode exhibited a TOF of 31.7 s^−1^ at a current density of 5.49 mA cm^−2^ and an overpotential of 200 mV for the CO_2_RR with water as a proton source. This value is only a direct estimate, as the CO_2_RR is much more complicated than this simple single reaction model, and a more detailed analysis is necessary for a more accurate TOF determination [41]. Comparatively, the device proved to be competitive with some catalyst materials found in the literature [42,43,44] within the parameters (solution, pH, etc.) used in this study.

## 4. Conclusions

In summary, the platinum electrode coated with poly-Pd(salen) exhibited electrocatalytic properties for the electroreduction reaction of CO_2_ in an aqueous solution. An important factor is the type of anionic counterion used in the electropolymerization process of the metal–salen complex. The hexafluorophosphate anion introduced molecular spaces of the appropriate size into the polymer to stabilize carbon dioxide, improving the efficiency of the coupling reaction. By Tafel analysis, this study highlighted that the CO_2_RR process on the metallopolymer occurred in three steps:(1)Reductive adsorption of carbon dioxide by the central metal.(2)Radial reduction of carbon dioxide to adsorbed CO.(3)Reduction of adsorbed CO to adsorbed -CHO.

The next studies will investigate the photoelectrocatalytic reduction of carbon dioxide.

## Data Availability

The data presented in this study are available on request from the corresponding author.

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
