# Peer review of "Electrocatalytic Reduction of CO2 in Water by a Palladium-Containing Metallopolymer"

_nanomaterials, 2022, doi:10.3390/nano12071193_

Round 1

Reviewer 1 Report

Teixeira  et al., studied CO2RR using Pd based Metallopolymer. The manuscript is well written. However, there several unclear points those must be resolved before publication.

Comments

  1. “The distribution of the particles over the electrode surface is controlled by the radius ion of the hexafluorophosphate anion”. The statement is not clear.
  2. Page 13: The authors claimed that “The Tafel slopes for these polarization regions were similar for both electrodes, and 222 region 2 was the rate-determining step for carbon dioxide reduction”. But, in Fig.5B. for hexafluorophosphate system Tafel slopes are presented as decimal of mV . It is quite low and seems unrealistic. Are the units V dec-1?
  3. Scheme 1: Three reduction steps are shown. Their potentials should also be reported in the scheme. What is the basis of the intermidiate and final product shown in the scheme?
  4. By RDE analysis, the authors have shown that reactions from -0.75 and -0.85 V is not diffusional. The reactions in this region perhaps adsorption controlled. The authors should record scan rate variant CVs and plot logI vs log (scan rate) plots in the said region to confirm the reaction is not diffusional. [ see Eq. 5-6 in literature; https://doi.org/10.1016/j.electacta.2021.139217 ]
  5. The authors used Pd based polymeric catalysts for CO2RR, however, how Pd and polymer helps the CO2RR is not clearly understood from the manuscript. Please clarify their roles.

Author Response

All amendments were taken into account in this revised version. The changes have been marked using YELLOW BACKGROUND. Regarding reviewer comments, firstly I thank for the interest and suggestions on the manuscript.

Reviewer 1

Comments:

1)  The distribution of the particles over the electrode surface is controlled by the radius ion of the hexafluorophosphate anion”. The statement is not clear.

Response:  According to the cited article (number 32). The distribution of metallopolymer particles is controlled by optimizing the transport of charge compensating ions. In the course of formation of a polymer, fragments of stacks acquire a positive charge, which is counterbalanced by anions of the supporting electrolyte. In an ideal situation, the stacks are arranged normally to the electrode surface at distances that are determined by dimensions of ions and solvent molecules.

To improve the understanding of the study, we have added text that describes this:

"During oxidative polymerization, the molecular columns acquire a positive charge and are normally arranged on the electrode surface at distances that are determined by the diameter of the anions and solvent molecules."

2) Page 13: The authors claimed that “The Tafel slopes for these polarization regions were similar for both electrodes, and region 2 was the rate-determining step for carbon dioxide reduction”. But, in Fig.5B. for hexafluorophosphate system Tafel slopes are presented as decimal of mV. It is quite low and seems unrealistic. Are the units V dec-1?

Response:  The reviewer is correct. The correct units are in V/dec. This correction was carried out in the graph.

3) Scheme 1: Three reduction steps are shown. Their potentials should also be reported in the scheme. What is the basis of the intermediate and final product shown in the scheme?

Response:  In Scheme 1, step information has been added. As in the legend containing the indications of the polarization regions observed in Fig 5 B.

4) By RDE analysis, the authors have shown that reactions from -0.75 and -0.85 V is not diffusional. The reactions in this region perhaps adsorption controlled. The authors should record scan rate variant CVs and plot logI vs log (scan rate) plots in the said region to confirm the reaction is not diffusional.

Response:  Thanks for pointing out the article. However, it is known that in an isothermal reaction it is not obeyed by the adsorption rate. In this way, the Levich application applies to the current under mass-transport-limited condition.  The linearity deviation of the polarization curve at different rotation rates indicates the potential dependence of the intrinsic kinetic current and consequently of the adsorbed species. This was proved by the Koutecky-Levich plot. Ref. Sensors 2002, 2, 473-506.

5) The authors used Pd based polymeric catalysts for CO2RR, however, how Pd and polymer helps the CO2RR is not clearly understood from the manuscript. Please clarify their roles.

Response:  We believe that in the course of the manuscript we have described the effect of metallopolymer on CO2RR. For example: “The onset overpotentials for dioxide carbon reduction were approximately -0.74 V and -0.46 V vs. SCE on the platinum electrode and poly-Pd(salen)-modified electrode, respectively. Based on the experimental results, we can see that the cathodic current increases for the modified electrode, requiring 0.15 V less overpotential than the platinum electrode to carbon dioxide electroreduction when considering the isocurrent for both electrodes. This decrease in overpotential is crucial for effective electrochemical reduction of CO2 by the catalyst. Moreover, the catalytic current of the modified electrode was 3.3 times greater than that of the unmodified platinum electrode at an isopotential of -0.95 V vs. SCE.”

Reviewer 2 Report

Design and fabrication of novel and high efficiency electrocatalytic materials are momentous and fundamental in the field of electrochemical reduction of CO2 to low molecular weight compounds. In this work, authors synthesized Pd(salen) metallopolymer on platinum-modified electrode by electropolymerization, and investigated the catalytic performance of metallopolymer for electrochemical reduction of CO2 by linear sweep voltammetry, Tafel analysis and rotating disk electrode measurements. Authors have interpreted their findings in a comparatively way, however, the deficiency of profound discussion and necessary characterization results can not be ignored. Unfortunately, it cannot be recommended for publication in present version in Nanomaterials. Attached with some detailed observations from this manuscript.   

(1) It is preferable to highlight the background, development status and advantages of metallopolymers in the field of electrochemical CO2 reduction reaction in Introduction part, as well as a brief elaboration of reason for selecting palladium.  

(2) It is preferable to add surface SEM image of unmodified platinum electrode, in order to correspond to AFM results.

(3) Authors claim “The counteranion imposes size and shape selection restriction through channels and pores readily adjusted in the polymer, exhibiting a molecular sieving effect. In comparison, the hexafluorophosphate anion introduced pores of the appropriate size into the polymer to stabilize carbon dioxide, improving the efficiency of the coupling reaction.” In Page 8, however, there is no porosity information or BET results in this work.

(4) What’s the nature of Pd(salen) metallopolymer? The XRD or TEM characterizations should be conducted.

(5) It is preferable to add more profound discussion on mechanism of enhanced catalytic performance in Results and Discussion part.

(6) Conclusion part should be reorganized or revised.

Author Response

The changes have been marked using YELLOW BACKGROUND. Regarding reviewer comments, firstly I thank for the interest and suggestions on the manuscript.

Reviewer 2

Comments:

1)  It is preferable to highlight the background, development status and advantages of metallopolymers in the field of electrochemical CO2 reduction reaction in Introduction part, as well as a brief elaboration of reason for selecting palladium.

Response:  We achieved this when we highlighted in the introduction the use of palladium for CO2 electrocatalysis, as well as the potential of the metallopolymer as an electrocatalyst. Below is our description in the introduction to our manuscript.

“The great challenge for catalysis using these metals is to increase the faradaic efficiency, to diminish the overpotential of catalysis and to ensure that the material is not poisoned by the formed byproducts.7-10 Huang and collaborators1 used a Pd icosahedra/C electrode and reached a faradaic catalytic efficiency of 91.1% for the selective conversion of CO2 to CO at a potential of -0.80 V vs. RHE. In another work, Kortlever et al.3 used carbon-supported bimetallic nanoparticles to create a Pt-Pd electrode and reached a faradaic efficiency of 88% for the conversion of CO2 to formic acid with a low overpotential (-0.40 V vs. RHE). These works demonstrate the catalytic potential of Pd-based materials in the search for better catalytic efficiencies and reduced catalytic overpotentials.

In the recent past, metallopolymers attracted much attention because of their excellent combination of redox-mediated switching capabilities, conductivity, and electrocatalytic properties.11-15 According to Elmas et al.16, metallopolymers exhibit a well-defined coordination environment, which can be applied in the development of smart materials with the redox properties of a transition metal cation. In electrocatalytic applications, metallopolymers function as hybrid materials capable of incorporating better catalytic properties than transition metals in conducting systems.17-21 In our research group, we have studied the electrocatalytic properties of metallopolymers containing salen moieties complexed with different metals in the application of fuel cells21, 22, chemiresistor sensors23 and electrochemical sensors24-26. The materials consist of an extended three-dimensional network formed by axial coordination of the d orbitals of the central metal with the π orbital of the phenolate moiety of the adjacent complex. Of these contributions, we have investigated in detail the effects of the different redox-active and charge transport properties of the metallopolymers.23, 27-29

2)   It is preferable to add surface SEM image of unmodified platinum electrode, in order to correspond to AFM results.

Response:  Images of conductive substrates are so common in the literature that we are more concerned with obtaining that of the proposed material. We won't have time to run a platinum SEM.

3)  In Page 8, however, there is no porosity information or BET results in this work.

Response:  According to the cited article (number 32). The distribution of metallopolymer particles is controlled by optimizing the transport of charge compensating ions. In the course of formation of a polymer, fragments of stacks acquire a positive charge, which is counterbalanced by anions of the supporting electrolyte. In an ideal situation, the stacks are arranged normally to the electrode surface at distances that are determined by dimensions of ions and solvent molecules. In fact, what we wanted to describe in the text is obtaining the molecular spaces between the metallopolymer columns. 

The text has been modified for better understanding.

4)  What’s the nature of Pd(salen) metallopolymer? The XRD or TEM characterizations should be conducted.

Response:  We would have liked to have carried out these studies. We know how important characterization of XRD and TEM is. However, our institution does not have equipment for such measurements. To accomplish them we would have to travel more than 500 km. In addition, research centers suspended activities during the pandemic.

5)  Conclusion part should be reorganized or revised.

Response:  ok

Reviewer 3 Report

Authors reported well-conducted study on the electrochemical reduction of carbon dioxide in mild conditions.

Authors did not fully successes in proving the novelty of this work upon the other present in literature. The must include an entire section where they compare the results achieved with those already published.

The electrochemical test were run in a very appropriated way and the y overall scientist soundness of the work is good enough. Authors should be improve the characterization of the electrode by including at least an XPS analysis on the fresh and used ones.

Furthermore, authors should check figure 3 captions. Did they use false colors?

Considering the points raised above, I cannot endorse the publication of this work in the present form. I can reconsider it after serous revisions and only if authors will be able to prove the novelty of this research.

Author Response

The changes have been marked using YELLOW BACKGROUND. Regarding reviewer comments, firstly I thank for the interest and suggestions on the manuscript.

Reviewer 3

Comments:

1)  The must include an entire section where they compare the results achieved with those already published.

Response: Compiling data from the literature and comparing them with our result is extremely difficult. This difficulty is that the different works do not use the same study conditions that we apply. Example: We could compare with articles that used palladium as an electrocatalyst for CO2, but some studied it in organic solvents, others studied metallic nanoparticles; etc... We believe that this comparison would be ideal for a review of the topic. However, in the manuscript text we have tried to demonstrate some significant electrochemical performance of the proposed material.

Such as: The onset overpotential for carbon dioxide reduction was approximately -0.22 V vs. NHE on poly-Pd(salen) modified electrode. In addition, by combining the electrochemical study with a kinetic study, the rate determining step of the electrochemical CO2 reduction reaction (CO2RR) was found to be the radial reduction of carbon dioxide to the CO adsorbed on the metal.

The counteranion used in the electropolymerization process has a direct influence on sta-bility during the formation of the radical cation and may, consequently, alter the network of the generated polymer and its electrochemical performance.

2)  Authors should be improve the characterization of the electrode by including at least an XPS analysis on the fresh and used ones.

Response: We would have liked to have performed an XPS study. However, our institution does not have equipment for this measurement. To carry out such measurements, we would have to travel more than 500 km. In addition, research centers suspended activities during the pandemic.

3)  Furthermore, authors should check figure 3 captions. Did they use false colors?

Response: Yes, the image has been artificially colored to better highlight the formation of molecular columns.  We added this information to the legend.

Round 2

Reviewer 1 Report

The authors revised the manuscript perfectly. Hence, it could now be accepted for publication.

Author Response

Dear Reviewiver 

Thanks for the comments. Regarding the review in English, I want to clarify that the manuscript before being submitted was edited for English language, grammar, punctuation, spelling by highly qualified native English editors of AJE (American Journal Experts). We forwarded a copy of the certificate of review to the editor. But even so, we carry out a review to check for spelling errors. 

Reviewer 2 Report

It may be accepted.

Author Response

Dear Reviewer

Thanks for the comments. 

Reviewer 3 Report

As i stated before if authors are not able to prove the innovation of tehir work there is not reason to submitt it to a peer review process.

Also, i understand the problems related to the pandemic eent but it cannot be an excuse to not perform an appropriated research. A research tha missed key studies cannot be published, and it is the case of this paper.

Authors did not improve the manuscript in a reasonable way. As it is, i cannot endorse its publication.

Author Response

Dear reviewer

We regret that you did not understand our situation in carrying out such measures required by the reviewer. And we disagree that we haven't made the necessary changes.

Regarding the review in English, I want to clarify that the manuscript before being submitted was edited for English language, grammar, punctuation, spelling by highly qualified native English editors of AJE (American Journal Experts). We forwarded a copy of the certificate of review to the editor. But even so, we carry out a review to check for spelling errors.
